# Prediction on Compressive and Split Tensile Strengths of GGBFS/FA Based GPC

**DOI:** 10.3390/ma12244198

**Published:** 2019-12-13

**Authors:** Songhee Lee, Sangmin Shin

**Affiliations:** Architectural Engineering, Graduate School, Chung-Ang University, Seoul 156-756, Korea; sh86136@naver.com

**Keywords:** GGBFS, compressive strength, split tensile strength, geopolymers

## Abstract

Based on rate constant concept, empirical models were presented for the predictions of age-dependent development of compressive and split tensile strengths of geopolymer concrete composite (GPCC) with fly ash (FA) blended with ground granulated blast furnace slag (GGBFS). The models were empirically developed based on a total of 180 cylindrical test results of GPCC. Six different independent factors comprising of curing temperature, the weight ratios of GGBFS/binder, the aggregate/binder, the alkali solution/binder, the Na_2_SiO_3_/NaOH, and the NaOH concentration were considered as the variables. The ANOVA analyses performed on Taguchi orthogonal arrays with six factors in three levels showed that the curing temperature and ratio of GGBFS to binder were the main contributing factors to the development of compressive strength. The models, functionalized with these contributing factors and equivalent age, reflect the level of activation energy of GPCC similar to that of ordinary Portland cement concrete (OPC) and a higher frequency of molecular collisions during the curing period at elevated temperature. The model predictions for compressive and split tensile strength showed good agreements with tested results.

## 1. Introduction

With mechanical and long-term material properties comparable to those of ordinary Portland cement concrete (OPC), along with its cost-effectiveness and environmental friendliness, fly ash-based geopolymer concrete (GPC) has become an attractive target to researchers as an alternative structural material to OPC [1,2,3,4,5,6,7]. Recently, geopolymer concrete composite (GPCC) has emerged with an alkali activated binder of a mixture of fly ash (FA) and ground granulated blast furnace slag (GGBFS) [8,9,10,11]. GPC blended with GGBFS enables the mixture to develop its compressive strength even at ambient temperature during the curing period [12,13,14].

Previous experimental results, mostly based on paste or mortar, showed that the development of the mechanical properties of geopolymer or its composite is influenced by various factors, which are listed in Table 1. A partial replacement of FA with GGBFS has also been observed to reduce the workability of GPC by a decrease in the setting time with the accelerated reaction, and by an increase in the amount of irregular shaped slag particles in contrast to spherical shaped FA [7,13,15,16,17,18,19,20].

As a limited number of studies have been conducted on the compressive and split tensile strengths of FA-based geopolymer concrete composite blended with GGBFS (GPCC) as well as their predictions, experimental investigations based on the Taguchi method were performed in this study in order to observe the effects of different factors on the fresh and hardened properties of GPCC. Based on these test results, empirical models predicting the compressive and split tensile strengths of GPCC at different ages were also presented.

## 2. Experiments

### 2.1. Materials

The base materials were low calcium FAs partially replaced with GGBFS. The chemical compositions of the FA and GGBFS determined through X-ray fluorescence (XRF) are given in Table 2. FA is classified as Class F in accordance with ASTM C618-12a (2012) [30]. FA and GGBFS were obtained from Sampyo Industry (Seoul, Korea) and Dangin Power Station (Dangin, Korea), respectively.

A combination of sodium hydroxide (NaOH) and sodium silicate solution (Na_2_SiO_3_), which manufactured from Samchun Chemicals (Seoul, Korea), was used as alkali activator. NaOH of 98% pure pellet was dissolved in tap water to produce NaOH solution. Commercially available Na_2_SiO_3_ solution was used. The chemical components of the Na_2_SiO_3_ solution were Na_2_O, SiO_2_, and H_2_O with corresponding weight percentages of 10.8%, 30.7%, and 58.5%, respectively. Crushed gravels with nominal maximum size of 19 mm with a specific gravity of 2.6 gr/cm^3^ and water absorptions of 1.5% were used. They were washed in order to minimize the effects of different relative levels of cleanliness of concrete properties in both the fresh and hardened states. River sand was used for the fine aggregates, which had a specific gravity of 2.3 gr/cm^3^, absorption of 2.7%, and fineness modulus of 2.2.

### 2.2. Determination of Levels of each Factor in the Taguchi Orthogonal Array

The main factors considered in different literatures were the curing temperature (T), the weight ratios of GGBFS/binder (r_GS_), the aggregate/binder (r_AG_), the alkali solution/binder (r_AS_), the Na_2_SiO_3_/NaOH (r_SH_), the concentration of NaOH in terms of its molarity (N_C_), the curing medium (water or steam), and the type of aluminosilicate source used (FA, GGBFS, or metakaolin). These factors and their ranges that have been considered in previous studies for both mortar and concrete as well as for composites, although the value of each factor ranged widely (Table 1). With reference to Table 1, six factors of T, r_GS_, r_AG_, r_AS_, r_SH_, and N_C_ were selected as the controlling factors of GPCC considered in this study. In the present experiments, the Taguchi design method was utilized with each factor being tested at three different levels (low, medium, and high), which resulted in 18 orthogonal arrays ($L_{18}\left(3^{6}\right)$) [31].

Table 3 illustrates the $L_{18}\left(3^{6}\right)$ orthogonal array of the Taguchi experiment with 18 trial mixes. Each sample was nominated as $Sn_{l_{4},l_{5},l_{6}}^{l_{1},l_{2},l_{3}}$ (or Sn in abbreviation), where S and n stand for sample and serial number in the $L_{18}\left(3^{6}\right)$ array, respectively, and $l_{1}$ through $l_{6}$ represent three levels (1, 2, and 3 for low, medium, and high, respectively) corresponding to the factors of T, r_GS_, r_AG_, r_AS_, r_SH_, and N_C_, respectively.

### 2.3. Fabrication and Testing of Specimens

A total of 180 cylinders (108 cylinders of 100 mm × 200 mm for compression tests and 72 cylinders of 150 mm × 300 mm for spit tensile tests) were fabricated. For each trial mix out of 18 sample mixes of $L_{18}\left(3^{6}\right)$, a batch containing six 100 mm × 200 mm and four 150 mm × 300 mm cylinders was mixed: six 100 mm × 200 mm cylinders to separately measure the compressive strengths in pairs at curing ages of 1, 28, and 90 days, and four 150 mm × 300 mm cylinders to separately measure the split tensile strengths in pairs at curing ages of 28 and 90 days. Mixing and specimen preparation were carried out at room temperature and relative humidity of 20 ± 2 °C and 50% ± 5%, respectively. Before mixing, the alkali activator was prepared by blending NaOH and Na_2_SiO_3_ solutions. These were stirred for 20 min. by stainless rod until viscosity disappeared. The steel cylindrical molds were then cleaned and brushed with a thin film of form oil in advance in order to prevent the agglutinate phenomenon between mold and concrete.

A 100-liter capacity concrete pan mixer was used for mixing each batch. For uniform dispersions of solid constituents, a total of 4 min. preliminary dry mixing was done without adding alkaline solution. The required amounts of coarse and fine aggregates were first put into the mixer and mixed for 1 min., then FA and GGBFS were added to the mix in sequence and an additional 3 min of dry mix was done. After the preliminary dry mix, alkaline solution was poured into the pan mixer and mixed for an additional 1 min.

The slump was measured immediately after the completion of mixing according to the standard test method specified in ASTM C143-05 (2005) [32]. The GPCC was cast into cylindrical steel molds and an electric oven was used to cure the specimens at the desired temperatures. To avoid water evaporation during curing, cylinders were sealed with the plastic sheets. All specimens were immediately taken out from the oven after 24 h curing. The compressive strengths of these specimens were tested at 1 day. The remaining eight specimens (four of 100 mm × 200 mm and four of 150 mm × 300 mm) were further cured at an ambient temperature of 20 °C until the additional tests were performed at 28 and 90 days to measure the compressive and split tensile strengths in accordance with ASTM C39 (2018) [33] and ASTM C496 (2017) [34], respectively.

## 3. Experimental Results and Effects of Factors

### 3.1. Overall Trends

The measured slump values were listed in Table 3. Figure 1 shows the measured slump values with respect to the molar ratios of H_2_O/Na_2_O. The measured slump values increased with increases in the molar ratio of H_2_O/Na_2_O up to a ratio of approximately 12.0. For the specimens $S4_{1,1,3}^{2,2,2}$, $S7_{2,1,3}^{3,1,3},$ and $S17_{1,2,3}^{1,3,3}$ with H_2_O/Na_2_O less than 12.0 (Table 3), respectively, the measured slumps were less than or almost equal to 150 mm. In general, these results indicate that a sufficient slump of even over 200 mm could be expected if H_2_O/Na_2_O exceeded 12.0, except for $S16_{3,1,2}^{3,2,3}$. Therefore, the mixes of GPCC designed with H_2_O/Na_2_O ratios greater than 12 could be regarded as having reasonable workability for casting structural members [5,35,36]. This observation regarding slump corresponded approximately to the one made by Hardjito and Rangan [37], in which slump values greater than 200 mm were obtained for GPC when the ratio of H_2_O/Na_2_O exceeded 10. 

The measured values of compressive and split tensile strengths were listed in Table 4. In general, compressive strengths at 28 days ($f_{c,28}^{\prime }$) tended to increase further from those measured at 1 day ($f_{c,1}^{\prime })$, without a noticeable additional increase from $f_{c,28}^{\prime }$ to the compressive strength at 90 days ($f_{c,90}^{\prime }$). This is because a high percentage of CaO in GGBFS allowed the hydration process to occur continuously at ambient temperature until 28 days [4]. The greater value of about 60 MPa was the average compressive strengths of $S3_{3,3,3}^{3,3,1}$, $S5_{2,2,1}^{3,3,2}$, $S9_{1,3,2}^{2,3,3}$, $S10_{3,1,1}^{2,3,1}$, $S15_{16,3,1}^{3,2,2},$ and $S16_{3,1,2}^{3,2,3}$ with relatively higher levels of $r_{GS}$ and higher curing temperature. The smaller one of about 20 MPa was the average of $S1_{1,1,1}^{1,1,1}$, $S6_{3,3,2}^{1,1,2}$, $S8_{3,2,1}^{1,2,3}$, $S12_{2,3,3}^{1,2,1}$, and $S18_{3,1,2}^{2,1,3}$ with relatively smaller levels of r_GS_ and lower curing temperature (Table 3).

Figure 2a illustrates the effects of T by comparing the ratios of compressive strengths for cylinders cured at 20 °C to that at 20 °C. For the relative increment of strength at each ages according to curing temperature, each two specimens with different T, 20 °C and 60 °C, were considered, which have the same values of $r_{GS}$. Relative increment of strengths were obtained from division of average compressive strength at 20 °C and at 60 °C (Figure 2a). Relatively significant increases in relative strength were observed for all values of $r_{GS}$ with increases in T. The average increases were 2.96, 2.09, and 2.24 for $f_{c,1}^{\prime }$, $f_{c,28}^{\prime }$, and $f_{c,90}^{\prime }$, respectively, as T increased from 20 to 60 °C.

Similar comparisons were made in Figure 2b for the effects of r_GS_. The average increases in relative compressive strengths measured from nine sets of a pair of specimens having the same T were 2.34, 2.11, and 2.07 for $f_{c,1}^{\prime }$, $f_{c,28}^{\prime },$ and $f_{c,90}^{\prime }$, respectively, as r_GS_ increased from 0.2 to 0.5. This indicates that, as compared to the effects of an increase in T, an approximately comparable effect of an increase in r_GS_ would be expected on the increase in compressive strength of GPCC.

In general, the split tensile strength was observed to increase with increases in the compressive strength (Table 4). Similar to the compressive strength, less significant changes between split tensile strengths were observed at 28 day ($f_{st,28}^{\prime }$) and 90 day ($f_{st,90}^{\prime }$).

### 3.2. Contribution of Each Factor by ANOVA

Figure 3 presents the relative and accumulated contributions of six factors to the development of compressive strengths at different ages obtained from ANOVA analysis. Figure 3 shows that the development of compressive strength was mostly influenced by the accumulated effects of T and r_GS_: 88.9% (56.5% and 32.4% by T and r_GS_, respectively), 86.4% (43.0% and 43.4%), and 87.4% (48.9% and 38.5%) for $f_{c,1}^{\prime }$, $f_{c,28}^{\prime },$ and $f_{c,90}^{\prime }$, respectively. The greatest contribution of T was observed in the development of $f_{c,1}^{\prime }$ at early stage of curing, but the additional development of compressive strength at ambient temperature was almost equally contributed to by both T and r_GS_ at the later stages of development at $f_{c,28}^{\prime }$ and $f_{c,90}^{\prime }$. Geopolymerization of Al-Si compounds seemed to occur mostly under elevated temperature conditions in one day, and then more gradual buildup of hydrates of C-S-H and C-A-S-H reacting with CaO in GGBFS formed in subsequent curing periods at ambient temperature. However, the contributions of the remaining four factors were found to be marginal for all stages of curing.

## 4. Development of Predictive Equations for Compressive Strength of GPCC

In order to develop a model for the prediction of the compressive strength of GPCC, two different approaches were attempted based on multiple regression and the concept of rate constant. 

### 4.1. Multiple Regression

Three separate regression analyses were performed by Equation (1) for the measured values of $f_{c,1}^{\prime }$, $f_{c,28}^{\prime }$, and $f_{c,90}^{\prime }$ at different ages, respectively, with each factor being normalized by its medium value at level 2. Table 5 was tabulated all values of coefficient for Equation (1).
(1)$$f_{c,d}^{\prime }=\beta_{0,d}+\sum_{i=1}^{6}\beta_{i,d}\cdot F_{i}+\sum_{i=1}^{6}\sum_{j=1}^{6}\beta_{ij,d}\cdot F_{i}\cdot F_{j}.$$

The comparisons between the measured and predicted values of $f_{c,1}^{\prime }$, $f_{c,28}^{\prime },$ and $f_{c,90}^{\prime }$ by these separate equations from Equation (1) were presented in Figure 4a with the corresponding averages (μ) and standard deviations (σ) for the ratios of the predicted to measured values of compressive strength. They were 1.00 and 0.17, 1.00 and 0.21, and 1.00 and 0.26 for $f_{c,1}^{\prime }$, $f_{c,28}^{\prime },$ and $f_{c,90}^{\prime }$, respectively. The overall values of μ and σ for all tested specimens were 0.99 and 0.22, respectively. The values of the coefficients at all ages, obtained for the factors of T ($F_{1}\;\mathrm{and}\;F_{1}^{2})$ and $r_{GS}$ ($F_{2}\;\mathrm{and}\;F_{2}^{2}$) being greater than those of the remaining four factors and the interaction term between T and $r_{GS}$ ($F_{1}\cdot F_{2})$, indicate that the T and $\;r_{GS}$ are the most highly-contributing factors but that their interaction effect is rather insignificant. Although the multiple regression model did show the overall trends of the contributions from each factor, no clear time-dependent relationships could be observed between the coefficients obtained for $f_{c,1}^{\prime }$, $f_{c,28}^{\prime },$ and $f_{c,90}^{\prime }$. As a result, there was no unified equation available, which could combine the three separate equations in Equation (1) into the one.

### 4.2. Rate Constant Concept

According to Bernhardt’s mathematical model [38] related to the rate of relative increases in concrete compressive strength with respect to its limiting compressive strength ($S_{u}$ in MPa), Tank and Carino (1991) [39] suggested Equation (4) to estimate the relative strength gain of OPC at equivalent ages of $t_{eq}$ (Equation (2)) and rate constant, $k_{r}$ (Equation (3)). Equation (4) was adopted in this study in order to predict the compressive strengths of GPCC at different $t_{eq}$s.
(2)$$t_{eq}=\sum_{o}^{t}e^{-E/R\left(\frac{1}{T_{c}+273}-\frac{1}{T_{r}+273}\right)}\cdot \Delta t,$$
(3)$$k_{r}\;=A\cdot e^{-E/\left[R\cdot \left(T_{c}+273\right)\right]},$$
(4)$$S=S_{u}\cdot \frac{k_{r}\cdot \left(t_{eq}-t_{0r}\right)}{1+k_{r}\cdot \left(t_{eq}-t_{0r}\right)},$$
where, S is the GPCC compressive strength at different $t_{eq}$s; $t_{eq}$ is the equivalent age (h)); $k_{r}$ is the rate constant at $T_{r}\;$($h^{-1})$; A is the frequency factor ($h^{-1}$); E is the activation energy in general (J/mol); R is the universial gas constant (=8.324 J/mol/K); r is the reaction coefficient (>0); $T_{c}$ is the temperature of concrete (°C); $T_{r}$ is the reference temperature (=20 °C in this study [40]); t is the real elapsed time (h); Δt is the time interval (h); and $t_{0r}$ is the age at the start of strength development at the reference temperature (h).

A total of 18 separate regression analyses were performed for 18 sample mixes of GPCC from $L_{18}\left(3^{6}\right)$ orthogonal array. Each regression analysis was performed to obtain the best-fitting values of A, E, $t_{0r},\;\mathrm{and}\;S_{u}$ in Equations (3) and (4) for the measured values of $f_{c,1}^{\prime }$, $f_{c,28}^{\prime }$, and $f_{c,90}^{\prime }$.

For OPC made of Portland Type I cement without admixtures or additives, the functionalized values of E with T or compressive strength were reported to be in the range of 4.0 × 10^4^–4.5 × 10^4^ J/mol [39,40,41,42,43,44,45]. For the GPCC considered in this study, a constant value of 4.0 × 10^4^ J/mol was obtained (Table 5), which is a close approximation of the value of E observed in OPC. This shows that a similar level of energy is required for GPCC to activate the initial chemical reactions as that of OPC. 

The pre-exponential factor (or frequency factor) of A in Equations (3) was found to be approximately $10^{5}/h$ for OPC [39,45,46,47]. For the GPCC used in this study, a constant value of $A=10^{6}/h$ was obtained from regression analyses for all sets, which is 10 times greater than the typical value of A observed for OPC. The predominant chemical reactions of geopolymerization in GPCC seemed to attribute to an increase in the frequency of the molecular collisions, which occur in a relatively shorter period of curing time at an elevated temperature in one day. 

The $t_{0r}$ in Equation (4), representing the age at the start of strength development, is in the range of 2.4–19.2 h for OPC at ambient temperature [39,45]. For the sample mixes of GPCC used in this study, an accelerated hardening of about 15 min. was observed for all cases regardless of the initial curing temperatures. A rapid reaction of GPCC reducing the setting time significantly was also reported by different researchers [13,48,49]. Regression analyses of the 18 trial mixes resulted in a value of $t_{0r}$ equal to zero, which reflects the rapid and accelerated nature of the strength development in GPCC.

Although consistent values of E, A, and $t_{or}$ were obtained for all 18 sets of the $L_{18}\left(3^{6}\right)$ orthogonal array, scattered values of $S_{u}$ between 18.8 and 66.0 MPa were obtained from the regression analyses. Accordingly, $S_{u}$ in Equation (4) was functionalized with two governing influential factors of $F_{1}$ and $F_{2}$ as given in Equation (1) based on the results obtained from previous ANOVA analysis.
(5)$$S_{u}=40.3\cdot F_{1}{}^{0.77}\cdot F_{2}{}^{0.63}\;\left(\mathrm{MPa}\right).$$

A unified model for the prediction of GPCC at all ages was obtained by substituting $S_{u}$ in Equation (5) into Equation (4), along with the values of A = 1.0 × 10^6^ ($h^{-1})$, E = 4.0 × 10^4^
(J/mol), and $t_{or}=0$ (h) obtained from regression analyses. Figure 4b shows the comparisons between the predicted and measured compressive strengths from a total of 108 measured specimens. The values of μ and σ for the ratios of predicted to measured compressive strengths of $f_{c,1}^{\prime }$, $f_{c,28}^{\prime }$, and $f_{c,90}^{\prime }$ were 1.09 and 0.21, 0.95 and 0.16, and 0.96 and 0.19, respectively. The overall μ and σ for all 108 specimens in 18 sets of $L_{18}\left(3^{6}\right)$ orthogonal array were 1.00 and 0.19, respectively. These statistical parameters obtained from Equation (3) were comparable to those reported for OPC in other studies [42,44,47]. The model based on the concept of rate constant concept (Equations (2)–(4)) could be regarded superior to multiple regression models with three separate equations (Equation (1)) by its better statistical parameters in spite of a unified single expression comprising strength development at different ages.

## 5. Prediction of Split Tensile Strength

In most structural codes and literature, the split tensile strengths of OPC and geopolymer-related mortar or concrete are typically expressed as a function of compressive strength in the form of Equation (4) [4,17,50,51,52].
(6)$$f_{st}^{\prime }=c_{1}\cdot {\left(f_{c}^{\prime }\right)}^{c_{2}},$$
where $f_{c}^{\prime }$ is the compressive strength of geopolymer concrete composite (MPa); and $c_{1}$ and $c_{2}$ are the empirical coefficients.

The final expression of S in Equation (4) was substituted into $f_{c}^{\prime }$ in Equation (6), and the best-fitting values of $c_{1}=0.47$ and $c_{2}=0.52$ in Equation (6) were obtained from regression analysis. In regression, a total of 72 split tensile strengths measured at 28 and 90 days were used as there were practically no difference in compressive strengths measured from 28 and 90 days. Figure 5a compares 72 measured values of $f_{st}^{\prime }$ with the predictions made by Equation (6). The values of μ and σ for the ratios of the model predictions to experimentally measured ones were 0.97 and 0.16, respectively. Figure 5b shows that $f_{st}^{\prime }$ of GPCC developed in this study is generally lower than that of OPC provided in different codes. The values of $f_{st}^{\prime }$ predicted for GPCC in this study varied from 0.82 to 1.02 and 0.89 to 0.91 times the values of $f_{st}^{\prime }$ of OPC predicted by the fib model code [51] and ACI 318-14 [52], respectively, as the compressive strength increased from 20 to 70 MPa. However, greater values of $f_{st}^{\prime }$ for GPCC considered in this study were predicted by 1.03 to 1.44, 1.04 to 1.07, and 1.11 to 1.14 times the predicted values of $f_{st}^{\prime }$ for GPC developed by Ryu et al. [17], GPC by Sofi et al. [4], and GPCC by Lee and Lee [50], respectively.

## 6. Conclusions

From the experimental investigations based on Taguchi’s 18 orthogonal arrays ($L_{18}\left(3^{6}\right)$), the following conclusions were drawn:

(1)The measured slump values tended to increase with increases in the molar ratio of H_2_O/Na_2_O. When the ratio of H_2_O/Na_2_O exceeded 12, a slump greater than 200 mm was obtained for most of the tested GPCCs.(2)ANOVA results indicated that T and r_GS_, among others, are more substantial contributing factors to the development of compressive strength of GPCC than the remaining four factors of r_AG_, r_AS_, r_SH_, and N_C_.(3)The development of compressive strength of GPCC was greatly affected by the early stage of curing at elevated T in one day, possibly due to the activated geopolymerization along with the hydration accompanied by a higher content of CaO in GGBFS.(4)A similar level of activation energy is required for GPCC and OPC. However, a higher frequency of molecular collisions could be expected during the chemical reactions in one day of curing at elevated temperature.(5)The unified model developed based on the rate constant concept with the limiting strength as a function of T and r_GS_ was able to predict the developed compressive strengths of GPCC at different ages with reasonable accuracy. Its overall statistical parameters were shown to be better than those from three separate multiple regression models obtained separately for different $f_{c,1}^{\prime }$, $f_{c,28}^{\prime },$ and $f_{c,90}^{\prime }$.(6)Using the predicted compressive strength by the rate constant model, an equation for the prediction of split tensile strength was also suggested. The predictions made for the split tensile strengths of GPCC tested in this study were shown to be about 10%–20% less than those of OPC, but greater than the predicted values for GPC or GPCC reported in other studies.(7)The developed GPCC may be used as structural concrete based on its mechanical properties and flowability comparable to those of OPC. The developed model based on the rate constant concept may be useful not only in determining the levels of influential factors for design favoring compressive or split tensile strengths, but also in predicting those at different ages.

## Figures and Tables

**Figure 1 materials-12-04198-f001:**
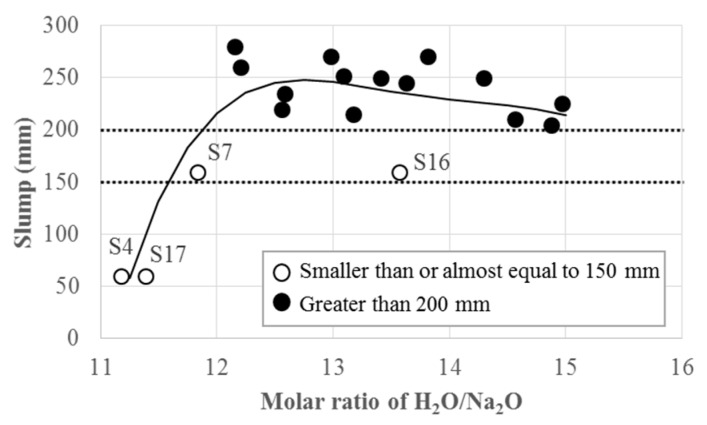
Measured slump values with respect to the molar ratios of $H_{2}O$/${\mathrm{Na}}_{2}O$.

**Figure 2 materials-12-04198-f002:**
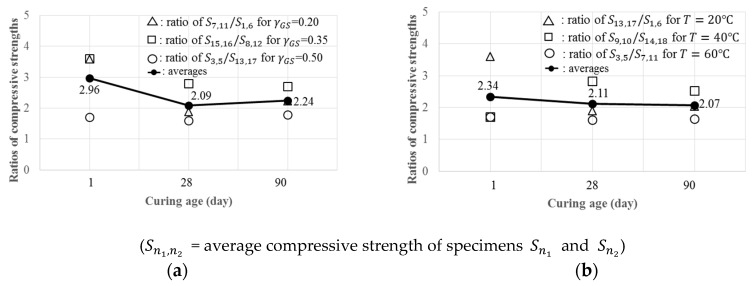
Effects of different factors on compressive strengths at different ages for the effect of T (**a**) and effect of r_GS_ (**b**).

**Figure 3 materials-12-04198-f003:**
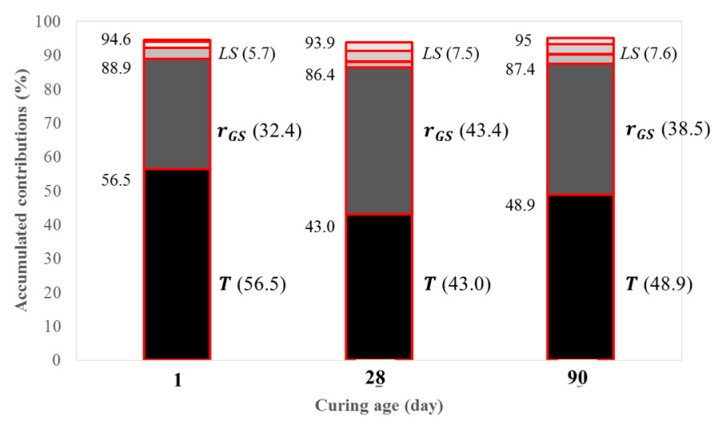
Relative and accumulated contributions of six factors to the development of compressive strengths at different ages obtained from ANOVA analyses (LS = lump sum effects of r_AG_, r_AS_, r_SH_, and N_C_).

**Figure 4 materials-12-04198-f004:**
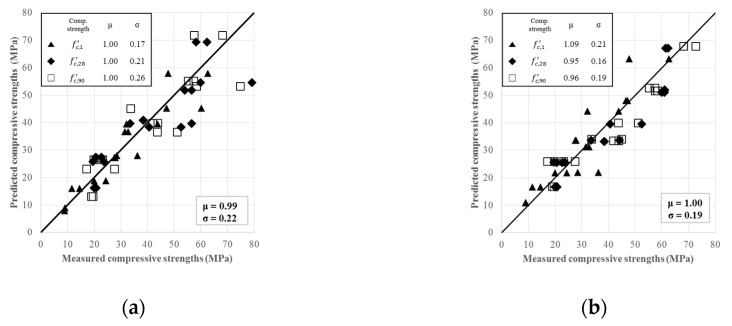
Comparisons between the measured and predicted compressive strengths for Equation (1) (**a**) and Equation (4) (**b**).

**Figure 5 materials-12-04198-f005:**
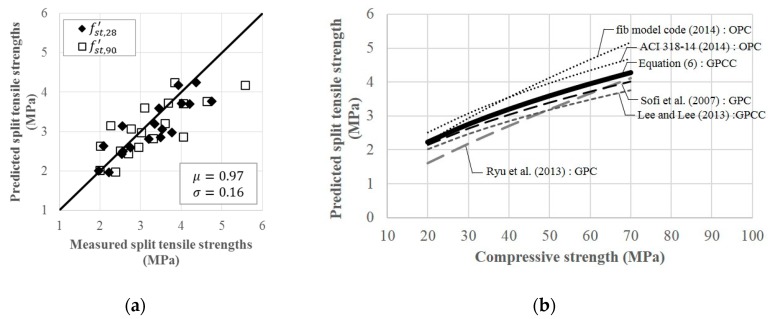
Split tensile strengths and their comparisons with Equation (6) and other predictions made for different types of concrete: comparisons between the 72 measured values of $f_{st}^{\prime }$ and their predictions by Equation (6) (**a**) and comparisons with other types of concrete (**b**).

**Table 1 materials-12-04198-t001:** Factors and their ranges considered in different literatures.

Factors	Units	Binders	Types	Min.	Max.
T [10,21,22,23]	°C	FA, GGBFS, Metakalin	GPCC, GP, GPC	25	90
r_GS_ [22]	-	FA, GGBFS	GPCC	0.30	0.55
r_AG_ [16,24]	-	FA	GPC	3.78	7.77
r_AS_ [24,25,26]	-	FA	GPC, GP	0.30	1.25
r_SH_ [13,24]	-	FA	GP	1	2.5
N_C_ [21,22,23,27]	M	FA, GGBFS	GPCC, GP	5	14
Curing medium [28]	-	FA, GGBFS	GPCC	-	-
Type of aluminosilicate source [27,29]	-	FA	GPC, GP	-	-

Note: curing temperature (T); the weight ratios of GGBFS/binder (r_GS_); the aggregate/binder (r_AG_); the alkali solution/binder (r_AS_); the Na_2_SiO_3_/NaOH (r_SH_); and the concentration of NaOH in terms of its molarity (N_C_).

**Table 2 materials-12-04198-t002:** Chemical compositions of fly ash (FA) and ground granulated blast furnace slag (GGBFS).

Parameter	Chemical Compositions (wt.%)										
	SiO_2_	Al_2_O_3_	Fe_2_O_3_	CaO	MgO	SO_3_	K_2_O	Na_2_O	P_2_O_5_	TiO_2_	LOI *
FA	57.42	22.0	8.35	5.80	0.99	0.39	1.71	0.45	1.08	1.35	0.68
GGBFS	32.77	13.68	0.44	44.1	3.05	4.06	0.48	0.27	0.03	0.69	2.39

* Loss on ignition.

**Table 3 materials-12-04198-t003:** Taguchi orthogonal array with 18 trial mixes ($L_{18}\left(3^{6}\right))$ and the measured slumps.

No.	Specimen Names	Main Factors						H_2_O/Na_2_O	Slumps (mm)
		T (°C)	r_GS_	r_AG_	r_AS_	r_SH_	N_C_ (M)		
1	$S1_{1,1,1}^{1,1,1}$	20	0.20	3	0.50	1.5	10	13.4	250
2	$S2_{2,2,2}^{2,2,1}$	40	0.35	3	0.55	2.0	12	13.1	252
3	$S3_{3,3,3}^{3,3,1}$	60	0.50	3	0.60	2.5	14	13.0	270
4	$S4_{1,1,3}^{2,2,2}$	40	0.35	3.5	0.50	1.5	14	11.2	60
5	$S5_{2,2,1}^{3,3,2}$	60	0.50	3.5	0.55	2.0	10	14.6	210
6	$S6_{3,3,2}^{1,1,2}$	20	0.20	3.5	0.60	2.5	12	13.6	245
7	$S7_{2,1,3}^{3,1,3}$	60	0.20	4	0.50	2.0	14	11.8	160
8	$S8_{3,2,1}^{1,2,3}$	20	0.35	4	0.55	2.5	10	14.9	205
9	$S9_{1,3,2}^{2,3,3}$	40	0.50	4	0.60	1.5	12	12.6	220
10	$S10_{3,1,1}^{2,3,1}$	40	0.50	3	0.50	2.5	10	15.0	225
11	$S11_{1,2,2}^{3,1,1}$	60	0.20	3	0.55	1.5	12	12.2	260
12	$S12_{2,3,3}^{1,2,1}$	20	0.35	3	0.60	2.0	14	12.2	280
13	$S13_{2,1,2}^{1,3,2}$	20	0.50	3.5	0.50	2.0	12	13.2	215
14	$S14_{3,2,3}^{2,1,2}$	40	0.20	3.5	0.55	2.5	14	12.6	235
15	$S15_{1,3,1}^{3,2,2}$	60	0.35	3.5	0.60	1.5	10	13.8	270
16	$S16_{3,1,2}^{3,2,3}$	60	0.35	4	0.50	2.5	12	13.6	160
17	$S17_{1,2,3}^{1,3,3}$	20	0.50	4	0.55	1.5	14	11.4	60
18	$S18_{2,3,1}^{2,1,3}$	40	0.20	4	0.60	2.0	10	14.3	250

**Table 4 materials-12-04198-t004:** Measured values of compressive and split tensile strengths.

Specimen Names	Compressive Strengths (MPa)									Split Tensile Strengths (MPa)					
	$f_{c,1}^{\prime }$			$f_{c,28}^{\prime }$			$f_{c,90}^{\prime }$			$f_{St,28}^{\prime }$			$f_{St,90}^{\prime }$		
	R1*	R2*	Avg.	R1	R2	Avg.	R1	R2	Avg.	R1	R2	Avg.	R1	R2	Avg.
$S1_{1,1,1}^{1,1,1}$	10.0	8.0	9.0	20.7	20.9	20.8	20.2	18.8	19.5	2.25	2.20	2.22	2.21	2.55	2.38
$S2_{2,2,2}^{2,2,1}$	28.1	27.6	27.9	53.3	51.8	52.6	56.0	46.1	51.1	2.42	2.67	2.55	2.25	2.26	2.26
$S3_{3,3,3}^{3,3,1}$	47.1	48.4	47.8	61.4	61.7	61.6	72.2	73.1	72.7	3.68	4.17	3.93	3.18	7.96	5.57
$S4_{1,1,3}^{2,2,2}$	34.4	20.9	27.7	40.6	40.7	40.7	43.7	43.7	43.7	3.62	3.07	3.35	3.60	3.60	3.60
$S5_{2,2,1}^{3,3,2}$	59.7	65.6	62.7	62.3	62.3	62.3	68.3	67.8	68.1	4.13	4.61	4.37	4.03	3.65	3.84
$S6_{3,3,2}^{1,1,2}$	9.4	8.5	9.0	18.6	21.5	20.1	19.4	18.6	19.0	1.93	2.01	1.97	2.18	1.80	1.99
$S7_{2,1,3}^{3,1,3}$	31.0	34.6	32.8	36.9	39.9	38.4	43.9	43.8	43.9	3.52	3.53	3.53	3.07	2.46	2.77
$S8_{3,2,1}^{1,2,3}$	16.1	12.8	14.5	25.3	20.1	22.7	21.4	24.9	23.2	2.71	2.42	2.57	2.39	2.60	2.50
$S9_{1,3,2}^{2,3,3}$	32.1	32.1	32.1	61.0	61.0	61.0	55.6	54.9	55.3	4.16	4.26	4.21	3.94	4.16	4.05
$S10_{3,1,1}^{2,3,1}$	43.3	44.3	43.8	62.8	59.6	61.2	53.6	61.0	57.3	3.69	3.21	3.45	3.15	3.04	3.10
$S11_{1,2,2}^{3,1,1}$	31.3	31.7	31.5	39.0	38.0	38.5	42.0	41.7	41.9	3.58	3.97	3.78	3.07	2.95	3.01
$S12_{2,3,3}^{1,2,1}$	11.8	11.3	11.6	19.1	22.1	20.6	18.1	21.4	19.8	2.28	2.79	2.54	2.93	2.46	2.70
$S13_{2,1,2}^{1,3,2}$	36.2	36.2	36.2	33.5	33.5	33.5	38.1	29.2	33.7	3.34	3.06	3.20	2.82	3.78	3.30
$S14_{3,2,3}^{2,1,2}$	24.3	24.4	24.4	24.2	23.1	23.7	26.3	28.7	27.5	2.53	2.96	2.75	2.84	3.06	2.95
$S15_{1,3,1}^{3,2,2}$	46.9	47.3	47.1	58.8	61.0	59.9	59.4	57.5	58.5	3.88	4.14	4.01	3.52	3.82	3.67
$S16_{3,1,2}^{3,2,3}$	45.8	47.2	46.5	60.4	61.4	60.9	59.7	55.4	57.6	4.68	4.81	4.75	4.60	4.67	4.64
$S17_{1,2,3}^{1,3,3}$	29.2	27.8	28.5	42.5	45.9	44.2	43.5	46.2	44.9	3.75	3.22	3.49	3.69	4.40	4.05
$S18_{2,3,1}^{2,1,3}$	19.5	20.4	20.0	21.2	17.8	19.5	18.6	15.6	17.1	2.15	2.01	2.08	2.16	1.85	2.01

Note: R1* and R2*: Two replicas of each specimen.

**Table 5 materials-12-04198-t005:** Specific coefficient for Equation (1).

Factors	Values of Coefficient for Equation (1)	
Dimensionless normalized factors ($F_{i}$)	$F_{1}=T/40,\;$$F_{2}=\;r_{GS}/0.35,\;$$F_{3}=r_{AS}/0.55,$ $F_{4}=r_{AG}/3.5,\;$$F_{5}=r_{SH}/2.0,$ and $F_{6}=N_{c}/12$.	
the best-fitting intercept and the coefficients of the normalized factors at different ages (d) ($\beta_{o,d}$, $\beta_{i,d}$, and $\beta_{ij,d}$)	d = 1 day	$\beta_{0,1}=4.2,{\;\beta }_{1,1}=2.7,{\;\beta }_{11,1}=1.8,$ $\beta_{2,1}=2.9,{\;\beta }_{22,1}=2.1,{\;\mathrm{and}\;\beta }_{12,1}=2.0$
	d = 28 day	$\beta_{11,28}=10.7,{\;\beta }_{2,28}=19.2,$ $\beta_{22,28}=2.7,{\;\mathrm{and}\;\beta }_{12,28}=5.8$
	d = 90 day	$\beta_{0,90}=1.7,{\;\beta }_{11,90}=13.0,{\;\beta }_{2,90}=9.3,$ $\beta_{22,90}=11.1{\;\mathrm{and}\;\beta }_{12,90}=1.7$
	All other β-values = 0	

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
