# Peer review of "Prediction on Compressive and Split Tensile Strengths of GGBFS/FA Based GPC"

_materials, 2019, doi:10.3390/ma12244198_

Round 1

Reviewer 1 Report

The reviewed paper presents empirical models for the predictions of compressive and split tensile strengths of geopolymer concrete composite. It is an interesting issue and this work provide novel experimental results, so it should be accepted. However, a revision is required based on the following comments:

Line 14: “ … with FA blended with GGBFS.”

When using an acronym in a text, the first time it is mentioned, the full name must be transcribed, followed in parentheses of the corresponding acronyms, so it should be written as: "...with fly ash (FA) blended with geopolymer concrete composite (GPCC)."

Line 22: “of OPC and …”

When an acronym is used in a text, the first time it is mentioned, the full name must be transcribed, followed by parentheses of the corresponding acronyms. Since this acronym is only used once in the summary, the full name must be used, so it should be written as: "... of Ordinary Portland cement concrete and ..."

Line 47: Table 1.

The acronyms T, rGS, rAG, rAS, rSH and Nc must be defined previous in the text (actually on line 66) or in a footnote to the Table 1.

Line 111. “The measured average values of compressive and split tensile strengths were..”

No data is provided to define the dispersion of the test results with respect to the average specimen values shown in table 4. 

Line 112. “In general, compressive strengths at 28 days (??,28′) tended to increase further from those measured at 1 day (??,1′), without a noticeable additional increase from ??,28′ to the compressive strength at 90 days (??,90′), “

This sentence ends with a comma. Replace it with an end point.

There are specimens that do not behaves as described, such as 13 and 16 among others, so Table 4 should be analysed more precisely.

Line 115. In the sentence that starts on line 115, it is recommended to write the full text, without using parentheses.

Line 123. Figure 1

It is not clear why specimens 4, 7, 16 and 17 are highlighted in Figure 1. The specimens 4 and 17 are mentioned in the text, but not the rest. It is recommended to label all the specimes as Sn, enough to know the composition.

Line 127. The sentence that starts on line 127 (“The average….”)

It should be specified the calculation method for increases of resistance at different temperatures, as well as the samples used for this purpose.

Line 169: the paragraph that starts on line 169.

I recommend to introduce a new table to show more clearly the values presented in the paragraph that starts on line 169.

Line 196: ecuation (1)

“S” is not defined.

Line 198:

teq should be presented as a new equation as well as Tr.

Line 201:”… temperature (=20 ℃ in this study [31]; …”

Close the parentheses before the semicolon.

Line 274: Figure 5b

Label lines do not correctly indicate curves.

Author Response

The authors sincerely appreciate the reviewer for his/her knowledgeable comments, which have helped the authors revise the manuscript. According to the reviewer’s comments, the authors corrected their manuscript.

Reviewer 2 Report

The manuscript presents the compressive and split tensile strengths and the slumps of fly ash‑based geopolymer concrete and the prediction of these properties. Before the publishing I would recommend following points to the attention of the authors:

Page 2, line 50 – 61: Who was the supplier of FA, GGBFS, Na2SiO3 and NaOH? Page 3, line 92-93: Did you avoid the water evaporation during the curing? Page 4: What caused the decrease of compressive strengths at 90 days in comparison with 28 days’ strengths of some samples? Page 5, line 137: The figures did not correlate with the caption under these figures. Page 5, line 143: Which software was used for ANOVA analysis? Page 8, line 274: It is not clear from the figure 5b which line represents which type of concrete.

Author Response

(The authors gave the same response as above.)

Reviewer 3 Report

In this paper, empirical models were presented for the predictions of age-dependent development of compressive and split tensile strengths of geopolymer concrete composite (GPCC) with FA blended with ground granulated blast furnace slag (GGBFS) were proposed.

The topic discussed is interesting and significant. However, there are a few issues that need to be addressed.

The abstract should indicate that GGBFS is ground granulated blast furnace slag and OPC is Ordinary Portland cement concrete.

In the introduction:

Line 37: after "...are listed in table 1" The following text should be added "This factor are: temperature (T), the weight ratios of GGBFS/binder (rGS), the aggregate/binder (rAG), the alkali solution/binder (rAS), the Na2SiO3/ NaOH (rSH); the concentration of NaOH in terms of its molarity (Nc), the curing medium (water or steam), and the type of aluminosilicate source used.

In the "Experiments" section, please explain in more detail the first paragraph (lines 85-90) of the subsection "2.3 Fabrication and testing of specimens".

Author Response

(The authors gave the same response as above.)
